# Design of a Low-Power and Low-Area 8-Bit Flash ADC Using a Double-Tail Comparator on 180 nm CMOS Process

**DOI:** 10.3390/s23010076

**Published:** 2022-12-21

**Authors:** Hong-Hai Thai, Cong-Kha Pham, Duc-Hung Le

**Affiliations:** 1Faculty of Electronics and Telecommunications, The University of Science, Vietnam National University, Ho Chi Minh City 700000, Vietnam; 2Department of Computer and Network Engineering, The University of Electro-Communications, Tokyo 182-8585, Japan

**Keywords:** MUX, flash ADC, TIQ, Double-Tail, comparator, encoder, SPI, CMOS

## Abstract

This paper presents a low-area 8-bit flash ADC that consumes low power. The flash ADC includes four main blocks—an analog multiplexer (MUX), a comparator, an encoder, and an SPI (Serial Peripheral Interface) block. The MUX allows the selection between eight analog inputs. The comparator block contains a TIQ (Threshold Inverter Quantization) comparator, a control circuit, and a proposed architecture of a Double-Tail (DT) comparator. The advantage of using the DT comparator is to reduce the number of comparators by half, which helps reduce the design area. The SPI block can provide a simple way for the ADC to interface with microcontrollers. This mixed-signal circuitry is designed and simulated using 180 nm CMOS technology. The 8-bit flash ADC only employs 128 comparators. The applied input clock is 80 MHz, with the input voltage ranging from 0.6 V to 1.8 V. The comparator block outputs 127 bits of thermometer code and sends them to the encoder, which exports the seven least significant bits (LSB) of the binary code. The most significant bit (MSB) is decided by only one DT comparator. The design consumes 2.81 mW of power on average. The total area of the layout is 0.088 mm^2^. The figure of merit (FOM) is about 877 fJ/step. The research ends up with a fabricated chip with the design inserted into it.

## 1. Introduction

Designing ADCs (analog-to-digital converters) has always been a topic that never gets old in electrical engineering due to the numerous applications provided by ADCs. The main function of an ADC is to convert an analog signal, which is continuous-time and infinite, to a digital signal, which varies discretely with regard to time. The three main steps of the progress of signal conversion in an ADC are sampling, i.e., sampling the analog input signal at a fixed value; quantizing, i.e., converting an infinite, continuous-time (analog) value to a finite, discrete-time (digital) value, which means that quantizing will decide the binary level that appears to be nearest to the sampled input value; and encoding, i.e., converting that level of signal to binary code or digital value. To evaluate how well an ADC can operate, the two concepts that should be focused upon are sampling rate, which relates to how fast an ADC can convert analog to digital, and bit resolution, which presents how precise an ADC conversion can be [1].

There are five major types of ADCs in use today [1]. Depending on the system specifications or requirements, an ADC will be chosen for the design after considering its features and capabilities. The SAR (Successive Approximation) ADC has the most balanced speed and resolution. The delta-sigma (ΔΣ) ADC provides high dynamic performance and inherent anti-aliasing protection. The dual-slope ADC calculates by comparing run-up and run-down time (two slopes). The pipelined ADC is a modern power-saving and very fast ADC type. The flash ADC is the fastest type; it provides an instant conversion without latency. The higher the required resolution, the bigger and more power-hungry the flash ADC becomes, so the sampling rate must be reduced. Flash ADCs are used in various applications such as digital oscilloscopes, microwave measurements, fiber optics, RADAR detection, wideband radio, etc. [1]. According to these features, a flash ADC has always received the most focus in ADC research. It is the fastest and contains a simple parallel architecture. These advantages make flash ADC compatible and reliable in most systems and designs. However, when only considering higher speed, no one can do anything about the power consumption of high-speed devices, which should much lower for the best performance. Because of the need for low-power consumption in all the devices, factors that can be modified are the reduction in the area or the speed [2].

The heart of every flash ADC is a comparator circuit. Some noteworthy techniques have been used in designing this type of circuit. For instance, the continuous-time comparator in [3] adds an auxiliary branch to the conventional comparator to detect the flip point and to compensate for the power loss caused by this point’s tail current. A threshold inverter quantization (TIQ) is a fascinating technique and is very popular in comparator design. There are great choices for TIQ comparators and for how to select them to apply in flash ADCs, which can be found in [4]. Take articles [5,6,7,8] as examples, which all discuss this structure when constructing flash ADCs. This brings a great advantage in power and spatial efficiency. However, this structure is sensitive to process variation and also increases the design time as the feature size variation is a long and repetitive task. Another remarkable comparator architecture is the standard cell, which is introduced in [9]. This research states that using basic gates as comparators is possible instead of using power-hungry analog comparators. Additionally, the threshold voltage is internal to the comparator, which avoids the use of a resistor ladder. On the other hand, the architecture relies less on linear circuits and migrates the design to the digital domain.

The proposed flash ADC in this paper comprises two significant parts: the comparator block and the encoder block. The comparator block compares and converts the analog input into a thermometer code. As mentioned above, there are various types of comparators whose functions differ from each other, regarding specifications or design requirements. This design applies a DT comparator architecture, which was generally introduced and improved in [10,11,12]. The target was to reduce the number of comparators by half compared to the conventional flash ADC. The presented design [13] generates the most significant bit (MSB) with a switched reference voltage. With just a single comparator, the MSB is decided upon to depend on the input signal. To help save power and make the design less complex, this architecture also implements the TIQ technique. The encoder block inputs the thermometer code, which is sent from the comparator block and converts it to a digital value as an output signal. The block processes the thermometer code in such a way that it becomes Gray code and then binary code. Both blocks are designed on 180 nm CMOS technology. Moreover, there is an 8:1 analog multiplexer and SPI block inserted to make the design more convenient while in use.

The key objective of this work is to build an 8-bit flash ADC with fewer resources compared to the conventional design to achieve low-power and low-area requirements. This should resolve the inherent weakness of a flash ADC. To achieve this, the ADC employs only half the number of comparators used in the conventional design. The DT structure is applied for all comparators in the design since it is a very low-power architecture. The design operates at a sampling rate of about 20–80 MHz with an input of 0.6–1.8 V. The target of low power was reached at 2.81 mW (80 MHz). The area used for the layout was 0.088 mm^2^, which also meet the low-area expectations. The FOM value is acceptable, as it ends up at 877.47 fJ/step. The design can be applied in small data-acquisition systems for future use.

The paper includes five main sections. The first one is this introduction section, which has given some overview regarding the design. Section 2 discusses the details of every main component that appears in this 8-bit flash ADC. After that, post-layout simulation results are presented in Section 3 to prove whether the design works fine or not. Section 4 provides measurements attained from the design and compares the design to other studies. Finally, the last section wraps up by providing some conclusions.

## 2. Design and Implementation

An overview block diagram of this 8-bit flash ADC is presented in Figure 1. The 8:1 analog multiplexer (MUX) selects between 8 analog inputs. Then, the selected analog signal *A_OUT* is sent to two main blocks—the comparator block and the encoder. The MSB of the 8 binary bits, which is *Q[7]*, is decided right after the analog input *A_IN* passes through a DT comparator. The remaining output bits, which are *Q[6:0]*, are decided by the thermometer code *T[127:1]* that is generated by the Comparator block, and then goes through the Thermo-to-Gray-to-Binary Encoder. To make the design more convenient for testing and use, we attach an SPI block that works as a slave. This would help the ADC transfer digital data adequately.

### 2.1. 8:1 Analog MUX

Figure 2 provides the schematic view of the 8:1 analog MUX. This MUX has 8 input channels which are 8 analog signals. There are 3 bits used for channel selection. Depending on these 3 bits, 1 of 8 analog signals will be picked and sent to the Comparator block. Each input line is followed by 3 serially connected NMOSs and 3 serially connected PMOSs. Each selection bit is divided into two separate lines with opposite logic (by passing through an inverter). Specific conditions with corresponding outputs of this MUX are presented in Table 1.

### 2.2. Comparator Block

In a flash ADC, the most significant part is the Comparator block. Requirements should be met regarding power and resource-saving but must make sure the ADC works a high speed. After researching various types of comparator architecture, it was decided that this paper would follow the comparator structure presented in [13]. A general view of a full N-bit flash ADC is suggested in Figure 3. 

As we can see, the leverage given by this architecture savis resources by reducing the number of comparators by half. For instance, conventional N-bit flash ADC requires $2^{N}-1$ comparators. However, this structure only needs $2^{N-1}$. This means an 8-bit flash ADC can be deployed with just 128 ($2^{8-1}$) comparators instead of 255, as is usual. We will take a deep look at the structure to get to know how this reduction is doable. Figure 4 presents a detailed schematic view of this work’s Comparator Block (referencing the design mentioned in [13]), which contains a TIQ Comparator, a control circuit, a 7-bit conventional flash ADC, and only one comparator for deciding MSB.

#### 2.2.1. TIQ Comparator

The TIQ comparator decides the controlling voltage for the (N-1)-bit flash ADC. It outputs one of two values *VDDA* or *GNDA* and sends that to the Control circuit. The structure comprises 3 stages of inverters connected in series as shown in Figure 5. The first inverter (or a single-input NAND gate) quantizes analog input (*IN*) based on the varying threshold voltage that is decided by the Width/Length ratio of PMOS and NMOS transistors. The two following inverters are used for increasing the gain as well as preventing unbalanced propagation delay.

#### 2.2.2. Control Circuit

The Control circuit, as shown in Figure 6, provides an auto-switched reference voltage to the resistors ladder based on the controlling input signal [13], which is generated by the TIQ comparator. The *VCTRL* input is generated from the above TIQ comparator output, which means this input serves only two voltage levels—HIGH (*VDDA*) and LOW (*GNDA*). The *VK* input is considered as a mid-point voltage, which is $\frac{\left(V_{in\left(\max\right)}+V_{in\left(min\right)}\right)}{2}$. The design contains a resistor divider between *VDDA* and *GNDA*. If analog input *IN* (of TIQ comparator) is higher than *VK*, *VCTRL* will be LOW and it will turn the two PMOS (M0 and M1) on. Then, *+VREF* charges up to *VDDA* = V_in(max)_ = 1.8 V, and *-VREF* charges up to *VK*. On the contrary, when *IN* is lower than *VK*, then *VCTRL* will be HIGH and it will turn M2 and M3 on. This makes *+VREF* equal to *VK* and *-VREF* equal to V_in(min)_ = 0.6V.

One thing that has to be considered here is that the series combination of resistors in the resistor ladder [13] is a big load applied between the two outputs. This makes the two decided reference voltage points (*+VREF* and *-VREF*) vary in an unwanted way. For example, the expected output value is between 1.2 V and 1.8 V, but with the load, the value is about 1.23 V to 1.78 V and would affect the comparison steps. Therefore, the M11 and M12 transistors are deployed for signal improvement and balance.

#### 2.2.3. DT Comparator

There are various types of comparator circuits chosen to deploy in the Comparator block. As we know, a good flash ADC requires consuming low power, optimizing in area, and performing at high speed. Therefore, this paper decides to use the DT comparator architecture that was suggested in [10] due to its great advantages. The proposed DT comparator, as shown in Figure 7, is separated into two stages—the below part is the *input stage* and the above part is the *output stage*.

There are a few notes about transistors in the design for a better understanding of the function. Mc1 and Mc2 are controlling transistors, and Msw1 and Msw2 are transistors that act as switches; we will get to know how they operate later. M1 and M2 are the input transistors, and M7, M9, M8, and M10 form back-to-back inverters which will decide the outputs. MR1 and MR2 are for separating stages and discharging output nodes to *GNDA*. This helps to save a lot of power.

In the beginning, when the clock input *CLK* is LOW (we can call this the “reset phase”), both Mtail1 and Mtail2 are OFF, and M3 and M4 charge the *FP* and *FN* nodes to *VDDA,* which makes MR1 and MR2 pull both outputs *OUTP* and *OUTN* to *GNDA*. In the “decision-making phase”, as *CLK* is HIGH, both Mtail1 and Mtail2 are ON, M3 and M4 are OFF, and the *FP* and *FN* nodes begin to drop. These nodes control MR1 and MR2 to decide output voltages. Suppose that the plus input *INP* is greater than the minus input *INN* (V_INP_ > V_INN_), and M2 conducts at a faster rate compared to M1. Therefore, *FN* reaches 0 V before *FP*. At the time *FN* drops, it turns Mc1 ON, to pull *FP* back to *VDDA*, and Msw1 OFF, to stop *FP* from falling to zero. That is how the design reserves power and reduces the pull-up time of those nodes. After *FP* and *FN* are decided clearly, they control the intermediate transistors—MR1 and MR2. As *FP* maintains at a HIGH level, MR1 is conducted and pulls *OUTN* down to *GNDA*. This leads to M8 opening and pulling *OUTP* up to *VDDA*. Outputs are decided as *OUTP* = 1, *OUTN* = 0 because of the input condition V_INP_ > V_INN_.

Below are some advantages of the above design that were also listed in [10]. The first thing is $\Delta V_{0}$, which is defined as the initial output voltage difference. The more this voltage increases, the sooner the outputs are decided. Additionally, Equation (1) below presents that $\Delta V_{0}$ rises by exponential growth:(1)$$\Delta V_{0}=4V_{thn}\left|V_{thp}\right|\frac{g_{mR1,2}}{I_{tail2}}\frac{g_{m1,2}\Delta V_{in}}{I_{tail1}}\;e^{\frac{G_{m,eff1}\cdot t_{0}}{C_{L,fn\left(p\right)}}},$$
where $V_{thn}$ and $V_{thp}$ are the threshold voltage of NMOS and PMOS transistors. The $g_{m}$ and $g_{m,eff}$ parameters are the transconductance (eff means effective) of the corresponding transistors. $C_{L,fn\left(p\right)}$ is the load capacitance of node *FP* or *FN*.

Secondly, with the pull-up to *VDDA* of one of the two nodes *FP* or *FN,* when the “decision-making phase” starts, which turns on MR1 or MR2, the output latching delay, $t_{latch}$ (since $\Delta V_{0}$ until the difference between two outputs is *VDD/2*), is reduced thanks to the effective transconductance’s increment ($g_{m,eff}$), as described in Equation (2):(2)$$t_{latch}=\frac{C_{Lout}}{g_{m,eff}+g_{mR1,2}}\cdot \ln\left(\frac{V_{DD}/2}{\Delta V_{0}}\right)$$

Finally, as mentioned above, the power consumed in this architecture is cut down to the minimum wherever possible.

Regarding the design issues [10], there are a few notes that should be considered. The control transistors (Mc1 and Mc2) must turn on as fast as possible before the regeneration begins. This can be achieved by using low-threshold PMOS devices. Regarding the NMOS switches (Msw1 and Msw2), we should apply large transistors to reduce the effect of voltage headroom limitation. The mismatch issue in this design includes threshold voltage mismatch and current factor mismatch. These depend on the ratio of the controlling transistor sizes to the input transistor sizes (Wc1,2/W1,2). This is because of the significant role of the input transistors as they amplify the input differential voltage (V_INP_ – V_INN_) so that they are usually deployed in large sizes, which helps prevent the mismatch of the two controlling transistors.

### 2.3. Encoder Block

Figure 8 presents the logic gate structure of the Encoder block. It is designed with the digital design approach. This block inputs the thermometer code, which was previously generated by the Comparator block, and converts it to a Gray code before translating it into Binary code as the final output. In this structure, the 127 input bits are named *T[1:127]*, which represents the thermometer code. After passing through multiple NOT, AND, and OR logic gates in parallel, the Gray code is generated as a 7-bit signal that is *G[0:6]*. Then, with XOR gates’ deployment, we obtain the 7-bit Binary code as the final result. Figure 8 also proves that each bit of thermometer code can affect only one bit of Gray code. This helps a lot in reducing the ability of noise appearance in signal and improving digital output’s accuracy.

### 2.4. SPI Block

For the *SPI block*, we refer to an open-source project in [14]. As the author showed, this is a simple Verilog SPI master/slave implementation featuring all 4 modes. The module is easy to use and understand with simple and flexible implementation. It features SPI master/slave support; all 4 modes (CPOL/CPHA); inverted data order support; and custom word size support. The *SPI block* works as an intermediary component to help users find it easy to read the flash ADC data by using a microprocessor (or microcontroller). This module contains an inverter, a clock divider, and—the main part—an SPI module. The inverter synchronizes with the entire system reset, since the SPI module is reset at the HIGH instead of the LOW level itself. The clock divider divides the SPI clock by 4. The most important part is the SPI module, which is set in slave mode. It receives in parallel the converted digital data from the Encoder, depicted by the *S_BINARY* pin. After 8 divided clock cycles, the data are shifted out serially through the *S_MISO* pin. The *S_MOSI* pin is left out since we only need to send the data out for further use.

## 3. Simulation Results

The proposed 8-bit flash ADC architecture is designed by using 180 nm CMOS technology. Each minor block should be tested separately before connecting it with other blocks. This helps to evaluate and to adjust the specifications of every block simply. The input range will vary from 0.6 VDC to 1.8 VDC. The reason it does not start from zero is because of the threshold voltage of transistors on 180 nm CMOS technology. Twenty- and eighty-MHz square wave clocks (*CLK*) are applied throughout the whole design. The post-layout simulation waveforms below are presented to prove how specific blocks function properly.

### 3.1. 8:1 Analog MUX

By applying eight different analog signals and running the selection bits throughout eight distinct conditions, we can check the analog mux function properly. The test provides eight sine waves with the same amplitude but at different frequencies from 1 to 8 kHz. The post-layout simulation result of the 8:1 analog MUX is demonstrated in Figure 9.

### 3.2. Comparator Block

#### 3.2.1. TIQ Comparator

The TIQ comparator is simulated in DC analysis as the input signal *IN* ranges from 0.6 VDC to 1.8 VDC. The output *OUT* results are shown in Figure 10. By adjusting the transistor sizes, the threshold voltage V_th_ will be decided as expected. The ideal requirement is that the output voltage meets the input at half of the input voltage range, which is V_th_ = 1.2 V.

#### 3.2.2. Control Circuit

The Control circuit uses two similar pairs of inverters. Therefore, the threshold level of those inverters must be ensured as we did previously with the TIQ comparator. The simulation result here does not include the load (resistor ladder), which will be deployed later in the whole system. The load narrows the output reference voltages, so additional M11 and M12 devices are added up to pull that range back to normal, as shown in Figure 11a. The input condition is that *IN* varies from 1.8 to 0.6 VDC. As we can observe, the output *VCTRL* from the previous TIQ comparator flips at mid-point 1.2 V from 0 V up to 1.8 V. Therefore, when *IN* is greater than 1.2 V, the [+VREF; -VREF] is [1.2 V; 1.8 V]. When *IN* is lower than 1.2 V, the [+VREF; -VREF] is [0.6 V; 1.2 V]. Figure 11b shows the DC simulation result, which provides a voltage transfer characteristics (VTC) curve [13]. This means that the two reference voltage +VREF and -VREF switch around mid-point voltage 1.2 VDC.

#### 3.2.3. DT Comparator

Figure 12a shows the simulation when applying the two inputs with a small difference in voltage value. The plus input (*INP*) stays at 704 mV in the first two clock cycles and then drops down to 696 mV in the next two clock cycles. The minus input (*INN*) is constant at 700 mV. The reason why the voltage difference is 4 mV is that it is lower than the ADC step size, which is $\Delta =\frac{1.8-0.6}{2^{8}}\approx$ 4.688 mV. This should prove that the comparator performs well even if there is the smallest difference between the two inputs. 

We obtained the result shown in Figure 12b by taking a closer look at a CLK cycle. We measured the discharge delay (t_0_) as a period starting from the rising edge of *CLK* until $\Delta V_{0}$ reached the transistor threshold at about 0.559 V. The result shows that t_0_ ≈ 4.1 ns. We waited a little more until the two outputs’ difference reached VDD/2 = 0.9 V, and we observedthe latching delay t_latch_ ≈ 200 ps. Therefore, the total delay t_delay_ ≈ 4.3 ns. In sum, right after the rising edge of the *CLK* signal, *FP* and *FN* take a small amount of time to decide. With such a close gap between the two differential inputs, the result is acceptable. This means the outputs will be decided even earlier in other cases if the gap is wider.

#### 3.2.4. Full Design of the Comparator Block

The entire Comparator block comprises the above parts—a TIQ comparator, a Control circuit, a DT comparator for MSB decision B[7], and 127 DT comparators for 127-bit thermometer code *T[127:1]*. There is an additional *Delay cell* whose purpose is to create a 5 ns (greater than the above t_delay_) delayed output *E_CLK*. It will be used as the Encoder block’s clock signal. This should help the Comparator block outputs have time to be stable before traveling to the Encoder block for further processing. The simulation results of the whole Comparator block are displayed in Figure 13 when the input voltage runs the full range from 0.6 V to 1.8 V. The results are separated into two figures for better observation.

According to 127 bits of thermometer code (*T[127:1]*), it is not possible to display all of those in a single-wave-view window. Hence, the simulation only presents some notable, equidistant bits, including *T[127]*, *T[96]*, *T[64]*, *T[32]*, and *T[1]*. The input runs from 0.6 to 1.2 V, as depicted in Figure 13a. In this part, the MSB of the output binary code *B[7]* stays at a LOW state since V_in_ < VK (1.2 V). In Figure 13b, with the input between 1.2 V and 1.8 V, the MSB bit is pulled up to the HIGH level. Both parts express the same thermometer code result, as it varies from 127 bits of zero (b’000…00) to 127 bits of one (b’111…11).

### 3.3. Encoder Block

An RTL was is written in Verilog HDL and then simulated and synthesized, its layout was drawn, and it was simulated post-layout with Synopsys’s EDA tools. The post-layout simulation result is presented in Figure 14.

As we can see, the 127-bit *T[127:1]* input is the thermometer code, and it runs from 1h (*0001) to 127h (*fff ffff ffff). The final result is exported as 7-bit *B[6:0]* output, and it ranges from 0d to 127d as expected. Each time the clock input *CLK* rises from LOW to HIGH (positive edge), the binary output is determined.

### 3.4. SPI Block

Figure 15 displays the post-layout simulation of the SPI block. The parallel input is 8-bit binary *B[7:0]*, which came from the Thermo-Gray-Binary Encoder and MSB comparator. The input master clock is the *S_CLK* signal, which should be at least 32 times the clock of the flash ADC. The data should be prepared before the chip-select *CS/SS* signal becomes LOW, and the data are shifted out serially. As we can see, the first 8-bit data b’11111111, the next one b’10101010, and the last one b’00000010 are all sent directly to the Master-in, Slave-out (*MISO*) line.

### 3.5. Full Design Simulation

Figure 16 presents the final design of this paper’s 8-bit flash ADC in full layout view, whose total area is 0.088 mm^2^.

The post-layout simulation result of the ADC is partially demonstrated in Figure 17. The output here varies from 56 to 76 as expected.

The testing condition is the ramp input *IN* running the full range from 0.6 to 1.8 V, which means the expected digital output will range from 0 to 255. The design had no problem converting the analog input to numeric values from 11 to 127 and from 132 to 255. However, errors occurred in some early values and some middle values. This should be noted in the future. The following analysis and measurement data will highlight the design problems, and some suggestions will be made to improve the quality of the design output in the future. Overall, approximately 94% of the numeric values (241/256) were correctly converted, which is functionally and operationally acceptable for a flash ADC design that has halved the area compared to other conventional designs.

## 4. Measurements and Comparison

### 4.1. Characteristics and Measurements

This design can operate at about 20–80 MHz clock/sampling rate. The power spectrum is demonstrated in Figure 18 with the input at a 19.531 kHz sine wave and a 20 MHz clock and in Figure 19 with input at a 78.125 kHz sine wave and a 80 MHz clock. The SFDR (Spurious Free Dynamic Range) at the 20 MHz clock is better than itself at 80 MHz, whose values are correspondingly specified as 31.44 dB and 38.49 dB. With the calculated ENOB (Effective Number Of Bits) value of 5.323 bits, the average power of 2.81 mW, and 80 MHz sampling frequency, the figure of merit (FOM) can be evaluated using Equation (3) below:(3)$$FOM=\frac{Power}{f_{sampling}\times 2^{ENOB}}\approx 877.47\left(fJ/step\right)$$

Table 2 provides some parameters and characteristics of this flash ADC.

### 4.2. Chip Fabrication

Figure 20a presents the real chip captured by a microscope with the flash ADC design in it. When zooming in, we obtained Figure 20b, which is a physical view of the design.

### 4.3. Summary and Comparison

The summary specifications of this high-speed 8-bit flash ADC are shown in Table 3, which also compares the results and measurement parameters of the design with ADC design from some other studies. The comparison focuses on the results of design area and power consumption.

Figure 21 provides an intuitive FOM versus sampling speed comparison of this design among some Flash/Folding ADCs and other types from ISSCC [20]. The graph also comprises the studies mentioned in Table 3. Although the sampling rate of this work is quite low, its FOM is better than many Flash/Folding ADCs from the survey.

## 5. Conclusions

In conclusion, the paper has presented a low-power and low-area 8-bit flash ADC on 180 nm CMOS technology. The design consumes 2.81 mW of power on average. The total area of the layout is 0.088 mm^2^. It presents an FOM of 877.47 fJ/step. The ADC was simulated and observed at 20 MHz and 80 MHz clock signals. The convertible input voltage was from 0.6 V to 1.8 V. The design consisted of two main functional blocks, the Comparator block, and the Encoder block. The Comparator block uses the DT comparator circuit architecture with low processing time as a base. At the same time, it is optimal in terms of the area when using only one DT comparator circuit to determine the MSB of the binary output. This reduces the number of comparator circuits by half compared to the conventional flash ADC. In addition, thanks to the optimization of the area, the design achieves low power requirements. The Encoder uses an architecture that switches from thermometer code to intermediate Gray code before converting to binary code output. The use of Gray code reduces the possibility of errors that affect the final digital data at the output. In addition, we also integrated an analog multiplexer to select the input analog signal channels—which can be applied to “pick up” input signals from data acquisition circuits such as Analog Front-End (AFE), and SPI communication block to transmit and receive data with microcontrollers or other peripherals in the future. Additionally, the load-driven blocks such as the TIQ comparator and the Control circuit could be improved to achieve higher specifications for further applications.

## Figures and Tables

**Figure 1 sensors-23-00076-f001:**
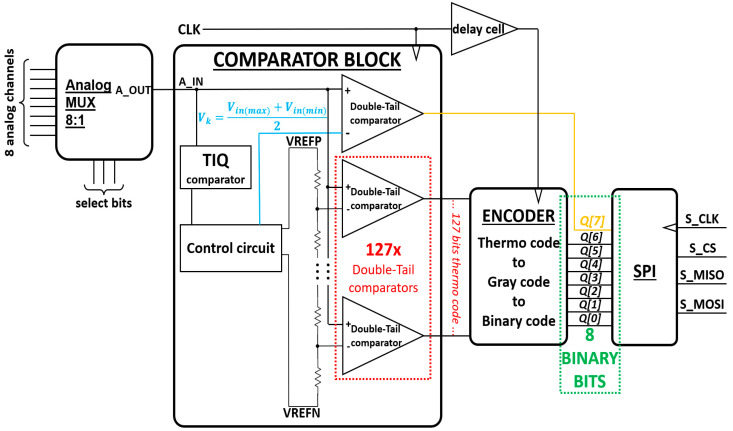
Block diagram of this 8-bit flash ADC.

**Figure 2 sensors-23-00076-f002:**
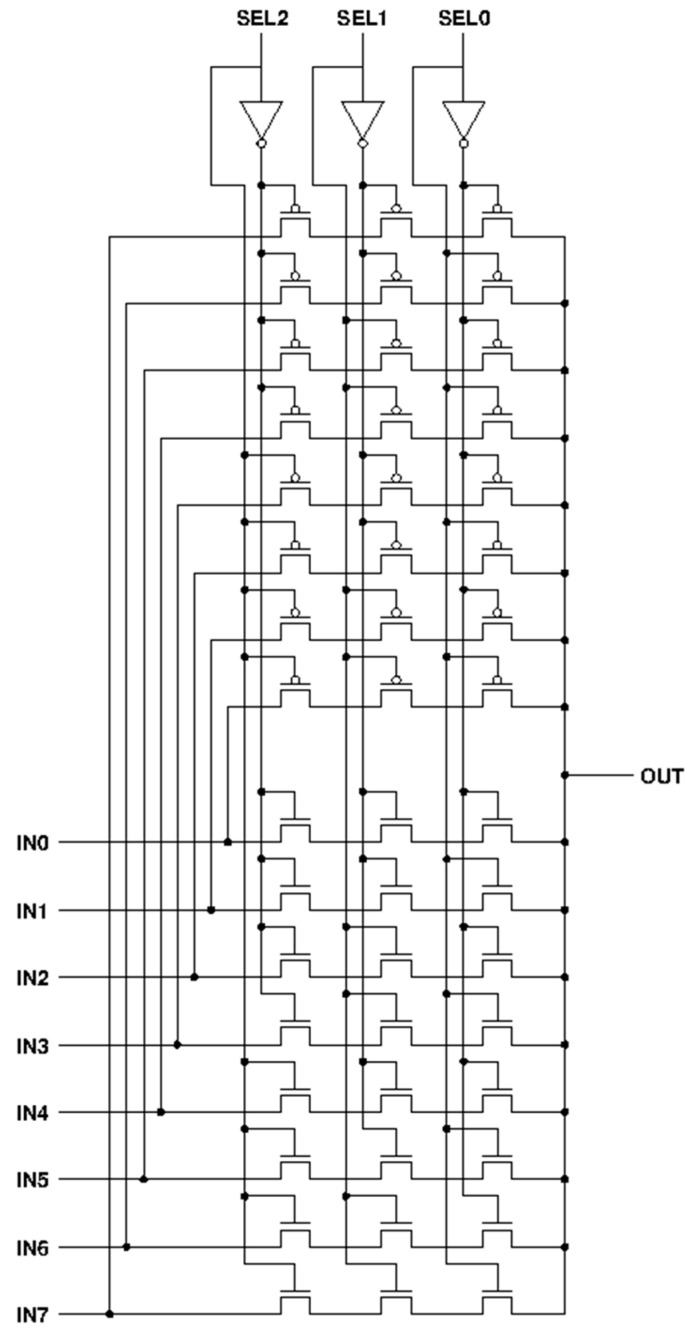
The 8:1 analog MUX.

**Figure 3 sensors-23-00076-f003:**
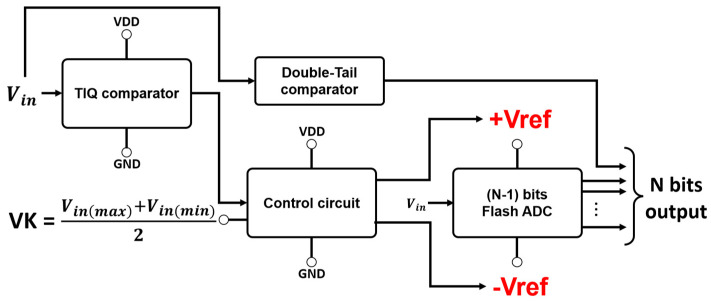
The proposed architecture of N-bit flash ADC.

**Figure 4 sensors-23-00076-f004:**
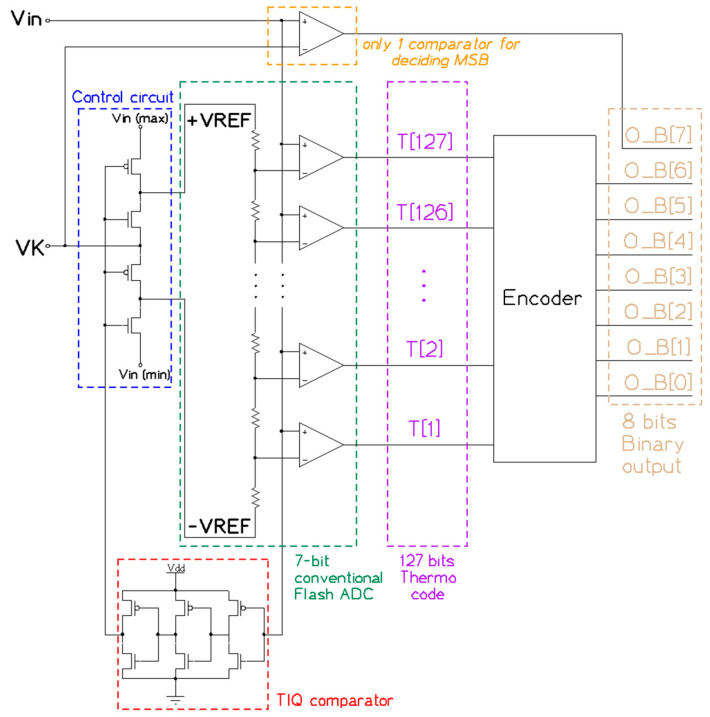
The detailed structure of the Comparator block.

**Figure 5 sensors-23-00076-f005:**
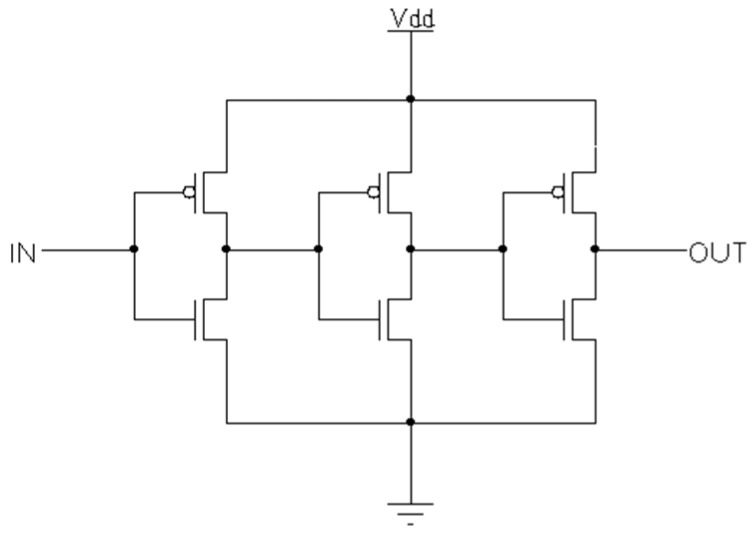
TIQ comparator.

**Figure 6 sensors-23-00076-f006:**
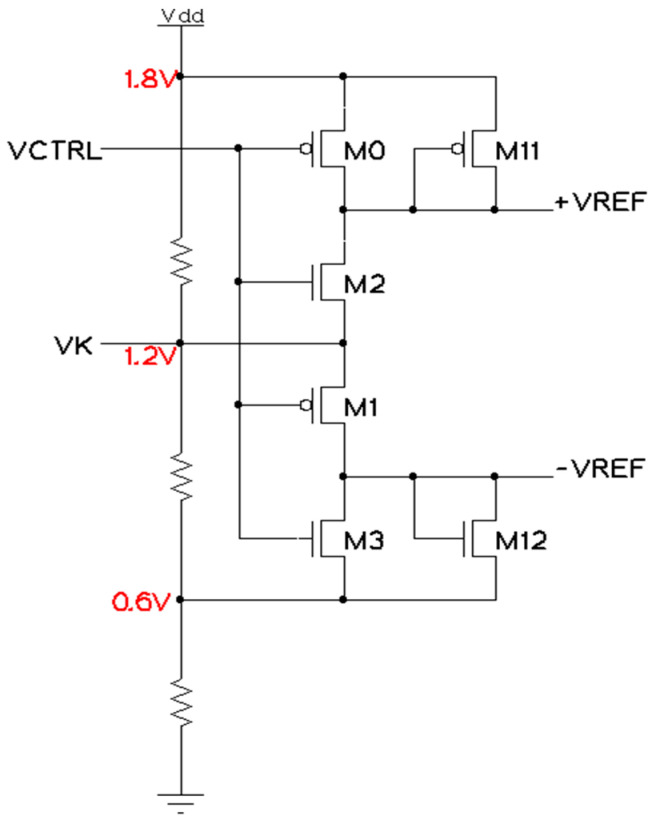
Control circuit.

**Figure 7 sensors-23-00076-f007:**
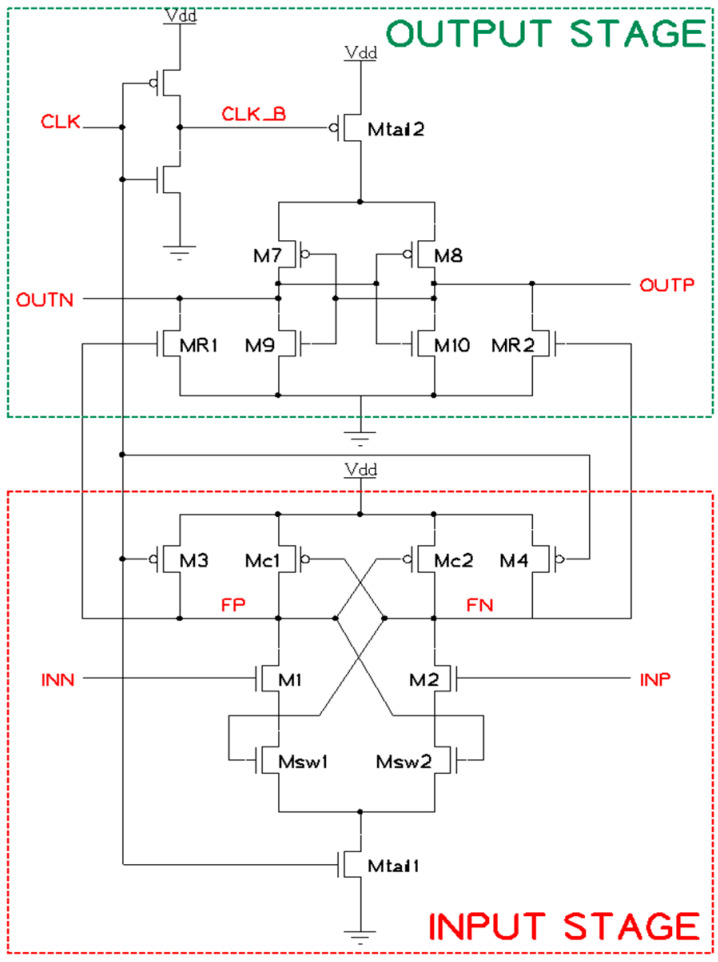
The proposed DT comparator.

**Figure 8 sensors-23-00076-f008:**
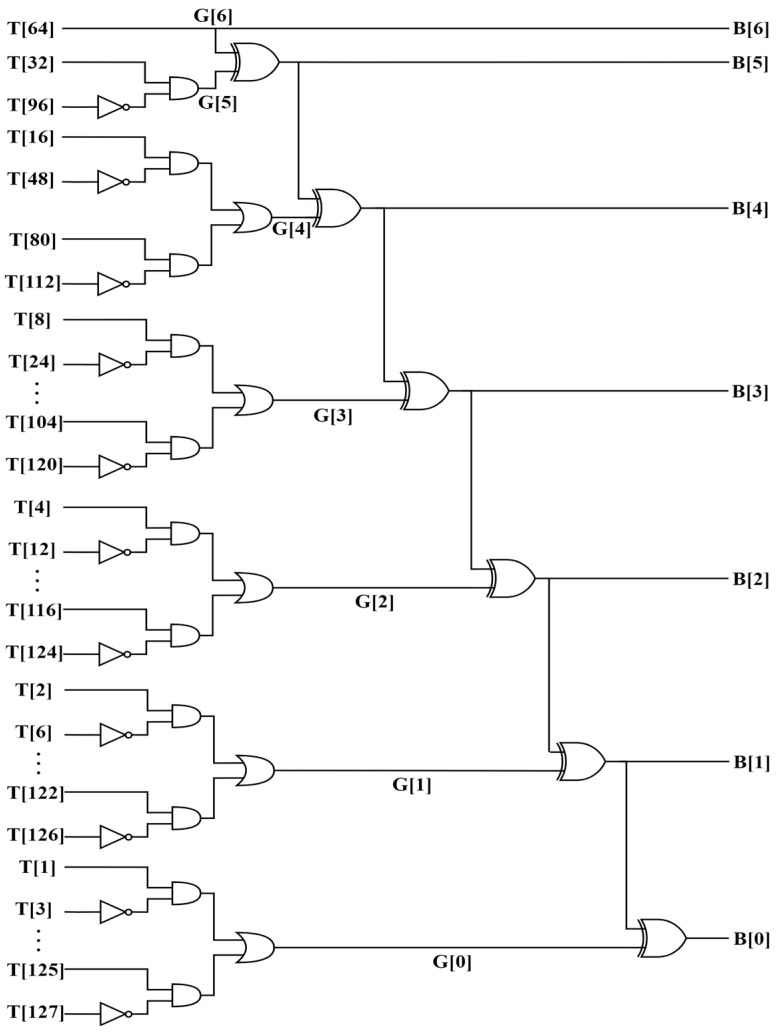
Encoder block.

**Figure 9 sensors-23-00076-f009:**
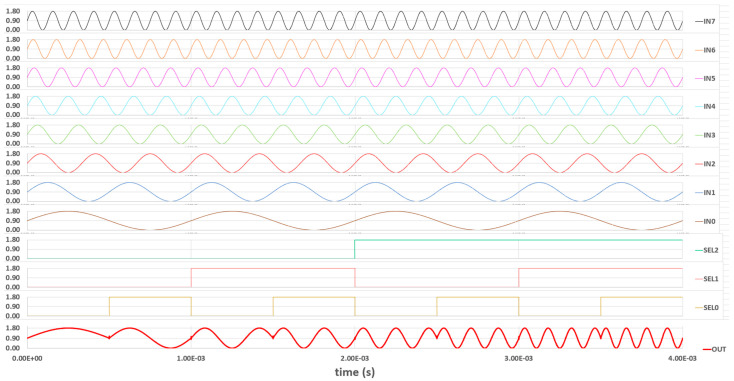
The 8:1 analog MUX post-layout simulation.

**Figure 10 sensors-23-00076-f010:**
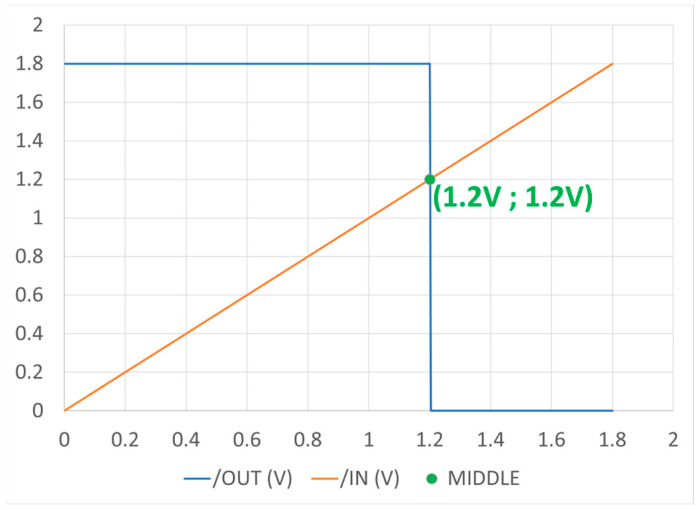
TIQ comparator post-layout simulation (DC).

**Figure 11 sensors-23-00076-f011:**
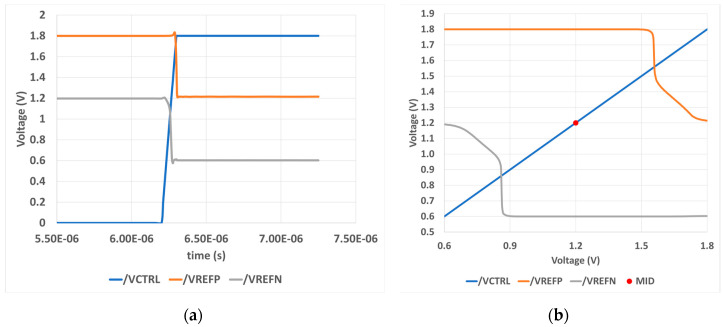
Control circuit post-layout simulation: (**a**) transient analysis; (**b**) DC analysis.

**Figure 12 sensors-23-00076-f012:**
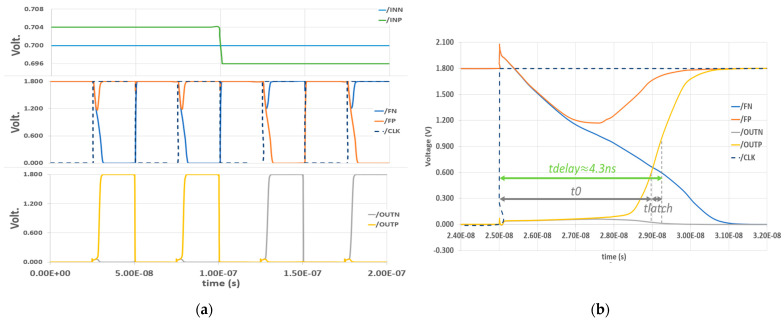
DT comparator post-layout simulation: (**a**) 4 clock cycles; (**b**) single clock cycle.

**Figure 13 sensors-23-00076-f013:**
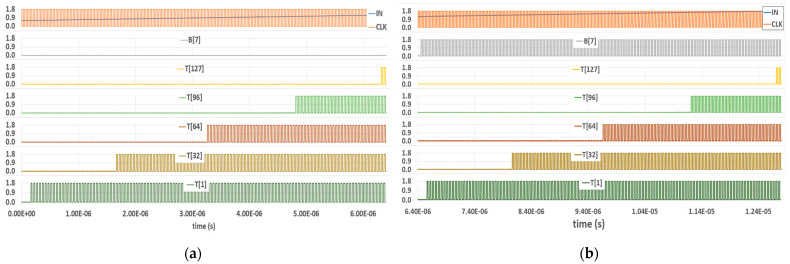
Comparator block post-layout simulation: (**a**) V_in_ = 0.6 to 1.2 V; (**b**) V_in_ = 1.2 to 1.8 V.

**Figure 14 sensors-23-00076-f014:**
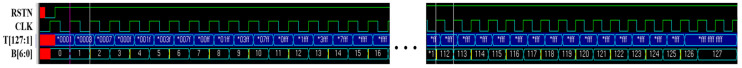
Encoder block post-layout simulation.

**Figure 15 sensors-23-00076-f015:**
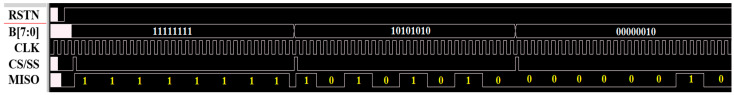
SPI block post-layout simulation.

**Figure 16 sensors-23-00076-f016:**
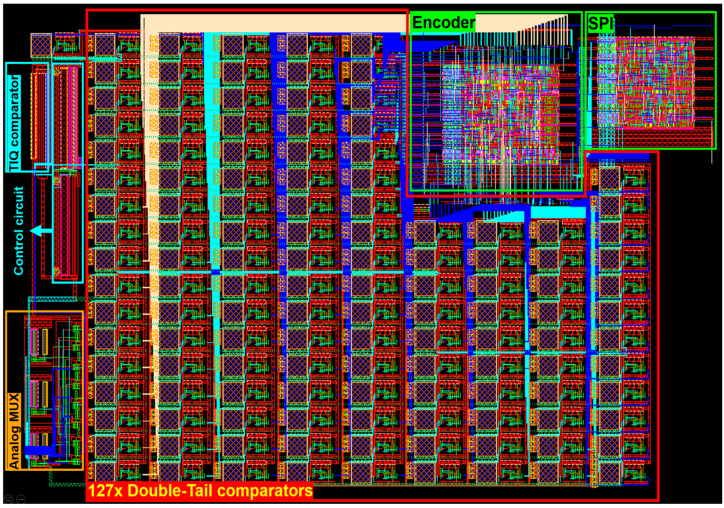
The layout of the flash ADC.

**Figure 17 sensors-23-00076-f017:**
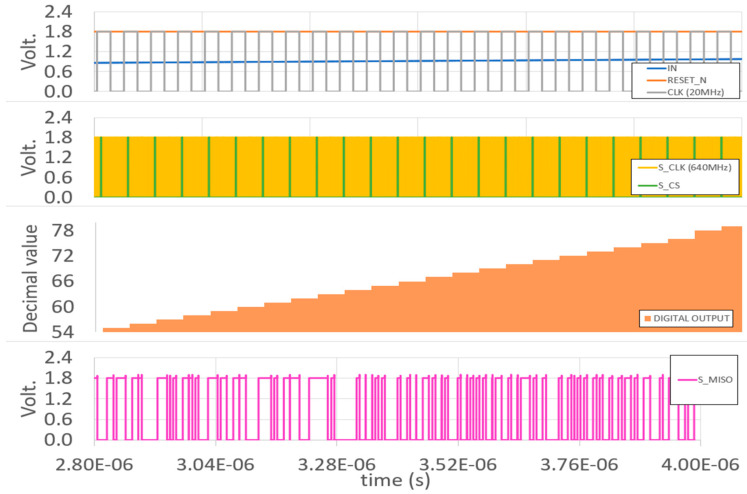
Full design post-layout simulation with output from 56 to 76.

**Figure 18 sensors-23-00076-f018:**
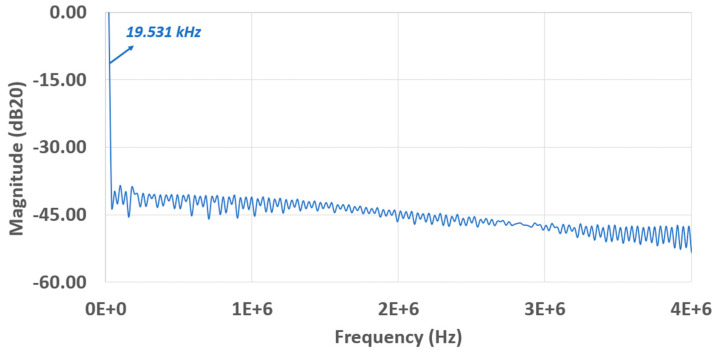
FFT spectrum with a 19.531 kHz input and a 20 MHz clock.

**Figure 19 sensors-23-00076-f019:**
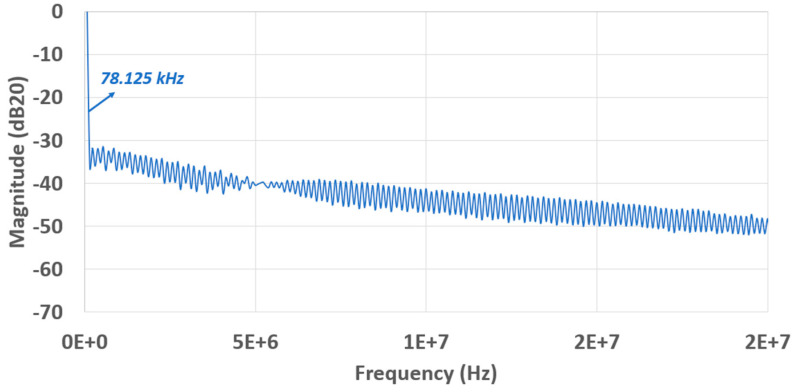
FFT spectrum with a 78.125 kHz input and a 80 MHz clock.

**Figure 20 sensors-23-00076-f020:**
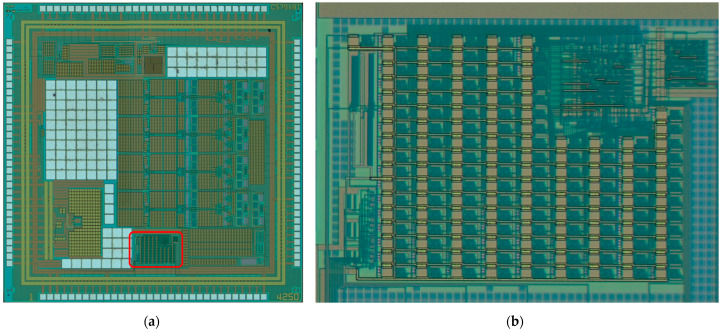
Chip fabrication result: (**a**) Real chip view; (**b**) microscopic view of the flash ADC only.

**Figure 21 sensors-23-00076-f021:**
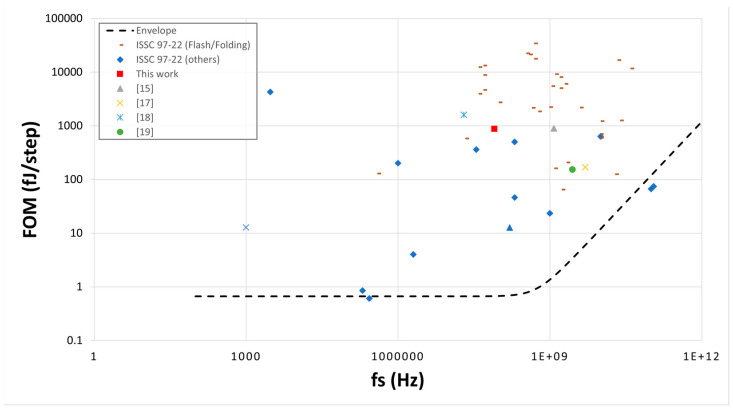
FOM vs. sampling rate.

**Table 1 sensors-23-00076-t001:** The truth table of the 8:1 analog MUX.

SEL2	SEL1	SEL0	OUT
0	0	0	IN0
0	0	1	IN1
0	1	0	IN2
0	1	1	IN3
1	0	0	IN4
1	0	1	IN5
1	1	0	IN6
1	1	1	IN7

**Table 2 sensors-23-00076-t002:** Characteristics and measurements.

	Value	Unit
INL (sigma)	0.591	LSB
DNL (sigma)	0.189	LSB
Sampling rate	80	MHz
ENOB	5.323	bit
SFDR	31.44	dB
Power	2.81	mW
FOM	877.47	fJ/step

**Table 3 sensors-23-00076-t003:** Summary and comparison with other ADCs.

	This Design	[15]	[16]	[17]	[18]	[19]
Process	180 nm	90 nm	180 nm	65 nm	180 nm	65 nm
Resolution	8-bit	6-bit	8-bit	5-bit	8-bit	8-bit
Supply	1.8 V	1.2 V	1.8 V	1.0 V	1.8 V	1.0 V
Samling rate	80 MHz	1.2 GHz	780 MHz	5 GHz	20 MHz	2.8 GHz
Area (mm^2^)	0.088	-	-	0.17	0.84	0.22
Power (mW)	2.81	21.1	53.0	21.0	4.64	51.0
FOM (fJ/step)	877.47	900	-	170	1600	153

## Data Availability

Data are contained within the article.

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
