# Peer review of "Design of a Low-Power and Low-Area 8-Bit Flash ADC Using a Double-Tail Comparator on 180 nm CMOS Process"

_sensors, 2022, doi:10.3390/s23010076_

Round 1

Reviewer 1 Report

The authors describe a design of an 8-bit Flash ADC in 180nm CMOS process. The paper is decently well-written and interesting to read. Nevertheless, it suffers from some serious flaws that would have to be remedied before publishing.

My major concern is the lack of the measurement results in terms of graphs. Yes, the authors provide a table, but in my opinion it would be a must to provide the actual measurement plots. Great that the authors have manufactured their design and proved that it works, but please, provide experimental data.

The other sever flaw of the paper is inappropriate comparison with state-of-the-art ADCs. Perhaps the presented design will be inferior, but it would be fine to publish even such a paper. Therefore, I suggest that the authors examine the results provided by the ADC performance survey: https://web.stanford.edu/~murmann/adcsurvey and put their design in such a context. Also, use designs from the survey in Table 3, comparison with other ADCs.

Regarding the presentation, please redraw schematics from Figures 2, 5, 6, and 7, in some specialized program for doing so, like XCircuit, instead of putting the screenshots from the schematic capture program. Please redraw Figure 4, since NMOS and PMOS transistors are barely distinguishable. Maybe you can use three-terminal symbols instead of the four-terminal symbols.

Why is the applied clock and sampling rate limited to 20MHz? Is that the maximum operating frequency (if yes why so low?) of the ADC or there is some other reason? Is the power shown in Tables 2 and 3 at 20MHz? If such a low, 20MHz sampling rate is used, why then not designing a SAR ADC instead which would be even more area and power efficient?

I assume you are using Walden FoM. If so, please define it in the paper.

Why are the encoder and the SPI slave controller separated and not synthesized and placed & routed together? I assume the area in that case would be lower.

Please read-proof your text. Also, please discuss the obtained results. Do not just write SFDR without even defining it.

Reviewer 2 Report

In this manuscript, Hai-Thai Hong et al proposed 8-bit Flash ADC architecture using Double-Tail Comparator on 180nm CMOS Process, which consumes low power and low area.

I believe the work could be published after some modifications which are described below.

 1. In line 9, please give the full name of the abbreviation SPI.

2. In line 73, the full name of the TIQ has been mentioned in line 56, there is no need to repeat it, the same situation appears in line 97, 98, 155, 160, etc.

3. In equation (1) and (2), please give the meanings of each parameter.

Reviewer 3 Report

Authors have presented work on "Design of a Low-power and Low-area 8-bit Flash ADC Using Double-Tail Comparator on 180nm CMOS Process". It is a good work. I have few suggestions / questions need to be addressed prior to accept it. 

1. I suggest to include key objective of the work with proper motivation in the last paragraph of Introduction section.

2. Include some recent literatures related to the work.

3. Include future scope of the study.

Round 2

Reviewer 1 Report

The authors have addressed my concerns as much as they could in such a short time. Finally, I would just request that the authors redraw all of their plots (figures 9, 10, 11, 12, 13, 17, 18, 19) since ticks on the x- and y-axis are barely visible. Please do not use screenshots from the tool itself, but export raw data points and redraw them in some specialized program for plot drawing.
